# A Comparative Study of the Efficacy of an Intervention with a Nutritional Supplement for Patients with Chronic Kidney Disease: A Randomized Trial

**DOI:** 10.3390/jcm11061647

**Published:** 2022-03-16

**Authors:** Sara Castro-Barquero, Marta Arias-Guillén, Sofia Pi-Oriol, Emilio Sacanella, Barbara Romano-Andrioni, Sandra Vidal-Lletjós, Ana María Ruiz-León, Ramon Estruch, Rosa Casas

**Affiliations:** 1Department of Medicine, Faculty of Medicine and Life Sciences, University of Barcelona, 08036 Barcelona, Spain; sara.castro@ub.edu (S.C.-B.); sofiapioriol@gmail.com (S.P.-O.); esacane@clinic.cat (E.S.); restruch@clinic.cat (R.E.); 2Department of Internal Medicine, Institut d’Investigacions Biomèdiques August Pi Sunyer (IDIBAPS), Hospital Clinic, 08036 Barcelona, Spain; amruiz@clinic.cat; 3Centro de Investigación Biomédica en Red Fisiopatología de la Obesidad y la Nutrición (CIBEROBN), Institute of Health Carlos III, 28029 Madrid, Spain; 4Department of Nephrology, Hospital Clínic Barcelona, 08036 Barcelona, Spain; marias@clinic.cat; 5Nutrition and Clinical Dietetics, Hospital Clinic de Barcelona, 08036 Barcelona, Spain; bromano@clinic.cat; 6Medical Department, Laboratoires Grand Fontaine, 08009 Barcelona, Spain; svidal@grandfontaine.eu; 7Mediterranean Diet Foundation, 08021 Barcelona, Spain

**Keywords:** malnutrition, chronic kidney disease, nutritional status, supplementation

## Abstract

Chronic kidney disease (CKD) involves heterogeneous diseases that affect the renal structure and function. Malnutrition plays a crucial role during patients with CKD on hemodialysis (HD) treatment and is associated with an increased rate and duration of hospitalizations. The aim of this randomized, parallel, intervention-controlled trial was to assess whether the use of daily supplementation with a new nutritional product developed by the Grand Fontaine Laboratories improves the nutritional status and anthropometric parameters of stage 5 CKD patients, compared with standard renal dietary advice, after three months of follow-up. Dietary intake, anthropometric measurements, physical activity, and blood samples were collected at baseline and after three months of intervention. Significant improvements were observed within the intervention group in body weight (1.5 kg [95% CI: 0.9 to 2.12 kg]) and BMI (0.54 kg/m^2^ [95% CI: 0.31 to 0.77]; *p*-value between groups, 0.002 and 0.006, respectively). In the control group, significant decreases were observed in transferrin saturation (−5.04% [95% CI: −8.88 to −1.21]) and alpha-tocopherol levels (−3.31 umol/L [95% CI: −6.30 to −0.32]). We concluded that daily dietary intake of a specific renal nutritional complement in CKD patients with or at risk of malnutrition may prevent deterioration in nutritional parameters.

## 1. Introduction

Chronic kidney disease (CKD) includes several heterogeneous conditions that affect renal structure and function [1]. In general, this disease is characterized by a progressive and irreversible deterioration of kidney function. CKD can be defined as a reduction in the glomerular filtration rate (GFR), increased urinary albumin excretion, or both. GFR is considered the main predictor of renal function, allowing for kidney status evaluation and classification into five stages [2]. In the last two decades, CKD has become a significant public health issue, with an estimated prevalence of 8–16% worldwide [3]. This indicates that the total number of adults that present with CKD is around 500 million, 225.7 million of whom are men (205.7–257.4 million), and 271.8 million are women (258.0–293.7 million) [4]. According the ENRICA (Estudio de Nutrición y Riesgo Cardiovascular en España) study, CKD affects approximately one in every seven adults in the Spanish population (15.1%), which is higher than that observed in the earlier EPIRCE (Estudio Epidemiológico de la Insuficiencia Renal en España) study (9.2%) [5]. The significant burden and costs of CKD on the healthcare system depend on two factors. First, dialysis and kidney transplantation, which are carried out in 1% of patients with CKD, are the most expensive treatments for chronic diseases. Second, there are high cardiovascular morbidity-mortality rates associated with this disease [5].

Malnutrition plays a crucial role in the evolution of CKD patients on hemodialysis (HD) treatment, and it is associated with an increased rate and duration of hospitalizations [6,7,8]. Various studies have estimated that 20–50% of HD patients are malnourished [9], which is a predictor of morbidity and mortality. Therefore, nutritional status should be periodically evaluated in CKD patients, and any nutritional deficiency must be adequately treated, especially for those undergoing HD [10]. Since there are some recommended dietary restrictions for CKD, particularly for protein intake [3,4,5], increased dietary protein may compromise the course of the disease [10]. Additionally, in CKD at the 5–5D stages, it may be necessary to limit fluid intake depending on whether residual urine is maintained, and, accordingly, the liquid intake recommendations should be individualized for each patient [6,7,8].

In this context, patients with renal failure, especially those at the CKD 3B–5D stages, are required to follow strict dietary recommendations that can sometimes lead to nutritional deficiencies. Accordingly, nutritional status should be periodically evaluated in these patients, especially in those undergoing HD [11], and they should be treated when necessary, beginning with nutrient enrichment and followed by oral nutritional supplements (ONSs). At present, the pharmaceutical market is deficient in products with moderate protein concentrations, and most of them do not have a satisfactory taste. Organoleptic and supplement formulation factors can lead to limitations in prescribing food complements to achieve the nutritional requirements for CKD patients.

The aim of the current study was to assess whether the use of daily supplementation with a new nutritional product developed by Grand Fontaine Laboratories improves the nutritional status and anthropometric parameters of stage 5D CKD patients, compared with standard dietary advice, after three months of intervention.

## 2. Materials and Methods

### 2.1. Study Design

A prospective, randomized, open, and controlled parallel-group dietary intervention trial was used to evaluate the safety and efficacy of the nutritional preparation FontActiv^®^ Renal HP during 3 months of intervention. 

### 2.2. Study Participants

From March 2019 to July 2020, 80 potential consecutive candidates from the Nephrology and Renal Transplant Department at the Center de Dialisi I Recerca Aplicada (DIRAC) and the Hospital Clinic de Barcelona were screened. However, 39 of these patients did not meet the inclusion criteria. The remaining 41 patients were randomized into the intervention and control (usual care) groups. Three patients withdrew from the study before the three-month endpoint (one due to death, one due to transplantation, and one at the request of a nephrologist). Thus, 38 subjects completed the study, with 19 in the intervention group and 19 in the control group.

The eligible participants were patients with stable chronic kidney failure in the last year (stage 5 CKD) who underwent HD or peritoneal dialysis (PD), without severe documented cardiovascular disease (ischemic heart disease, angina, myocardial infarction, cerebrovascular accident, peripheral vascular disease), and with malnutrition or high risk for malnutrition. Nutritional status was assessed in all of the participants, both at the beginning of the study and at the end of the intervention, using the subjective global assessment (SGA), which has been previously validated in CKD patients [12]. The SGA assesses changes in dietary intake and body weight, gastrointestinal disturbances, and renal functional decline [12].

Participants with a previous history of cardiovascular disease, any serious chronic disease, alcoholism or other drug addictions, or gastrointestinal diseases that present difficulties with following an adequate diet were excluded. Those who took vitamins, soy, or nutritional supplements during the month prior to the study were also excluded. Prior authorization from a nephrologist who followed up with the patient was mandatory, and all participants provided written informed consent to participate in the study.

### 2.3. Product to Test

FontActiv^®^ Renal HP is a ready-to-drink, nutritionally complete and balanced drink that is high in bioavailable milk proteins (16 g/200 mL bottle), enriched with dietary fiber, and high in energy (2 kcal/mL), that is designed for dietary management in patients with CKD. This preparation is also enriched with vitamins and minerals, does not contain lactose, and is gluten-free. In addition, this product is low in sodium, potassium, and phosphorus. The nutritional values for FontActiv^®^ Renal HP are described in Appendix A. According to the manufacturer’s instructions, this preparation should be shaken before being opened and it is recommended to be served cold. As a supplement, it is recommended to consume 1–3 bottles/day. As it is necessary to limit the intake of liquid in CKD, in this study, we recommended that patients consume 1 bottle/day. Regarding the organoleptic characteristics of the product, it has a vanilla flavor.

### 2.4. Intervention, Physical Activity, and Clinical Measurements 

Once the written informed consent was signed, participants were randomly assigned to two groups (nutritional intervention or control), while preserving gender equality among the intervention arms. Randomization was performed by consecutive allocation using sealed envelopes, which were made using a computer-generated random number table. All participants underwent an initial (baseline) examination and were then asked to maintain the assigned intervention for 3 months (15 weeks). A total of 41 participants were randomized, with 21 allocated to the nutritional intervention group (11 men and 10 women) who consumed one bottle (200 mL) of food supplement per day, and 20 participants (10 men and 10 women) who received standard nutritional advice based on the dietary recommendations for CKD. The use of the oral supplement was individualized to the nutritional requirements of each participant, considering their lifestyles and daily routines to guarantee supplement intake. No other dietary interventions were performed on these participants.

In order to minimize any possible discomfort (e.g., vomiting, feelings of fullness) associated with food complement intake, the participants were generally recommended to drink the product cold and, for those who were more intolerant, to divide it throughout the day. All participants received instructions on how the product should be consumed. To ensure compliance, the participants were asked to bring the used bottle caps from the product to each follow-up visit.

Participants in the control group, who did not consume the dietary supplement and showed some degree of malnutrition, received standard care. 

All participants had face-to-face interviews with a dietitian at baseline and after 3 months. Data on sex, age, educational level, sociodemographic characteristics, and clinical and lifestyle information were gathered. In addition, a validated 151-item food frequency questionnaire (FFQ) was used to quantify food intake in the previous year [13]. Before and after each follow-up visit, each participant also completed a brief questionnaire on the possible adverse effects of the dietary supplement. Physical activity was assessed using the short Minnesota Leisure Time Physical Activity questionnaire validated for the Spanish population [14].

After a dialysis session, anthropometric measurements, such as height, weight, and waist and abdominal circumference, were collected by trained dieticians at baseline and after 3 months. In addition, plicometry was applied to measure the triceps skinfold, and a measuring tape was used to measure arm and leg circumference. HD is usually performed three times per week, and the visits were made during the intermediate weekly session, thus ensuring fluid and electrolyte control. Body mass index (BMI) was calculated by dividing weight (kg) by the square of height (m) and applying the specific cutoff point for dialysis patients [15]. Due to the potential for hyperhydration, which is usually observed in patients undergoing HD or PD, body weight and BMI were assessed after dialysis sessions. Body weight was measured with the participants in light clothing without shoes or accessories, using a high-quality calibrated scale. Waist circumference was measured above the iliac crest, at the level of the hips, and on the most prominent part of the buttocks. The triceps skinfold was measured at the back side of the middle-upper arm, arm circumference was measured on the front side of the middle-upper arm, and arm and leg circumferences were measured at the largest point of the left arm and thigh, respectively.

### 2.5. Ethics Statement

The study protocol and procedures were approved by the Committee for Ethical Research (CEI) at the Hospital Clínic de Barcelona and are in agreement with the ethical standards of the Declaration of Helsinki. The study was registered with the following code: HCB/2018/0865.

### 2.6. Laboratory Measurements

Blood samples were collected at baseline and at the end of the intervention for the determination of biochemical and nutritional parameters, including hemoglobin, total lymphocytes, total proteins, albumin, prealbumin, transferrin, and protein bound to retinol. Routine analyses were performed at the CORE Laboratory at the Hospital Clinic de Barcelona, which fulfilled all required quality criteria, receiving ISO 9001:2015 certification and ISO 15189 accreditation for some tests. The levels of parathormone (PTH), C-reactive protein (CRP), transferrin saturation, serum, and intraerythrocytic folic acid, vitamins A, B1, B12, C, D, and E, zinc, magnesium, and selenium were also determined. In addition, other blood parameters, including red blood cell count, hematocrit, mean corpuscular volume, leukocytes and platelet counts, glucose, creatinine, electrolytes, uric acid, transaminases, lactate dehydrogenase, alkaline phosphatase, gamma-glutamyl transpeptidase (GGT), bilirubin, and the lipid profile (total cholesterol, HDL, LDL, triglycerides, ApoA1, Apo B and lipoprotein A) were determined. Blood samples were always collected before beginning the dialysis session (immediately in the morning or afternoon, depending on the shift).

In brief, the blood parameters were determined for each participant using frozen samples of whole serum or plasma, as appropriate. Blood glucose levels were determined using the glucose oxidase method, serum insulin levels by radioimmunoassay, cholesterol and triglyceride levels by enzymatic procedures, and HDL levels after precipitation with phosphotungstic acid and magnesium chloride. All of the preceding analyses were performed using an Advia 2400 clinical chemistry analyzer (Siemens Healthcare, Tarrytown, NJ, USA). In the case of the apolipoproteins A1 and B, the levels were measured using turbidimetry. Serum folic acid, homocysteine, and vitamin B12 levels were assessed using an automated electrochemiluminescence immunoassay system (Advia-Centaur, Siemens, Tarrytown, NJ, USA). Additional serum parameters, including plasma aspartate aminotransferases (AST), alanine transaminase (ALT), GGT, albumin, and prealbumin levels, were measured to determine the possible adverse effects of the dietary supplement.

### 2.7. Statistical Analysis

For a parallel design, the required sample size was determined using the ENE 3.0 statistical program (GlaxoSmithKline, Brentford, UK), assuming a maximum loss of 10% of participants. To detect a mean difference of 2% in the change in concentration in prealbumin and protein (malnutrition degree) with a conservative standard deviation (SD) of 2.66%, 19 subjects would be needed to complete the study (risk = 0.05, power = 0.9). Malnutrition parameter improvements were considered the primary outcome and were used to determine the sample size, but changes in all endpoints were of equal interest in this study. Differences in dietary intake, excluding ONSs, were assessed using one-way ANOVA and comparisons between groups after 3 months of intervention with paired *t*-test. Non-normally distributed variables were compared with Mann–Whitney U test. Differences in anthropometric and biochemical parameters at baseline and after 3 months were assessed using one-way analysis of variance, as appropriate. Changes after 3 months were assessed by ANCOVA adjusted for the baseline levels of each variable. Normality was assessed for all the variables using the Kolmogorov–Smirnov test. For non-normally distributed variables, differences between the groups were assessed using the Mann–Whitney–Wilcoxon test. Within- and between-group differences are expressed as estimated means and 95% CIs, while, for non-normally distributed variables, the differences are expressed as median (25th, 75th percentile). Categorical variables are expressed as percentages. The significance level was set at *p* < 0.05. All analyses were performed using SPSS v. 20.0 (SPSS Inc., Chicago, IL, USA).

## 3. Results

Of the 80 participants initially assessed for eligibility, 20 and 21 were finally randomized into each of the two groups. Figure 1 shows the retention rates (≥90% for all) at the 3-month of follow-up. One participant was lost in the control group by death, two in the intervention group by transplantation, and another at the request of a nephrologist.

It is worth highlighting that none of the participants received any type of supplementation during the study period except for the product under evaluation in the intervention group.

The baseline characteristics of the participants are shown in Table 1. The proportionality between men and women (50% per group) was ensured. No statistically significant biochemical differences were observed between the two groups. However, some anthropometric parameters, such as BMI, body weight, and arm circumference, were significantly different between groups (all *p* < 0.05). 

### 3.1. Tolerance of the Dietary Supplement

Tolerance of the dietary supplement was acceptable. Some side effects were reported during the study, which caused abandonment (n = 1, 4.8% of the 21 randomized participants) or difficulty in adhering to product intake in some patients (n = 3, 15.8% of the 19 participants who finished the intervention). The most common side effects were fullness, rough tongue, nausea and vomiting, dizziness, and stomach-swelling, generally occurring in patients older than 65 years. Patients younger than 65 years had fewer side effects and, therefore, greater adherence and clinically relevant anthropometric improvements.

### 3.2. Food, Energy Balance, and Dietary Adherence

No statistically significant differences were observed in dietary intake, excluding the ONS intake, at baseline and after 3 months of intervention between intervention arms, except for alcohol intake (*p* = 0.046) (Table 2). Significant differences were observed in control arm after 3 months of follow-up, with a reduction in SFA (*p* = 0.014), sugar (*p* = 0.040), vitamin C (*p* = 0.033) and sodium intake.

### 3.3. Anthropometric Measurements

The effects of the 3-month nutritional intervention on anthropometric parameters are detailed in Table 3. Significant baseline differences were observed between intervention arms in BMI and some anthropometric parameters. Significant improvements were observed after adjustment for baseline levels of each anthropometric measurement in the intervention group in body weight (1.5 kg [95% CI: 0.9 to 2.12 kg]) and BMI (0.54 kg/m^2^ [95% CI: 0.31 to 0.77]). In addition, significant differences between groups were observed in body weight and BMI (*p* = 0.002 and 0.006, respectively). No statistically significant differences were observed in the control arm or for the rest of the anthropometric parameters, but a positive trend was observed in the intervention arm for the triceps fold. Likewise, there was a tendency toward a decrease in the calf and arm circumference and triceps fold in the control group. No changes in physical activity were observed for either group (data are not shown).

### 3.4. Biochemical Parameters

Changes in the biochemical parameters are summarized in Table 4. Significant baseline differences were observed between groups for glycated hemoglobin. Significant increases were observed in the intervention group in Vitamin D (8.80 ng/dL [25th to 75th percentile: 5.80 to 11.4) and selenium levels (9.16 μg/dL [95% CI: 1.36 to 17.0]). In the control group, significant decreases were observed in transferrin saturation (−5.04% [95% CI: −8.88 to −1.21]) and alpha-tocopherol levels (−3.31 μmol/L [95% CI: −6.30 to −0.32]). Furthermore, significant differences were observed between groups in vitamin D and phosphorous levels (*p* = 0.010 and 0.045, respectively).

## 4. Discussion

Our results suggest that a nutritional intervention with an ONS can improve nutritional status and reduce the risk of protein-energy wasting (PEW) in CKD patients undergoing dialysis. Participants who followed the nutritional intervention increased their body weight, while the participants without nutritional support show a deteriorated nutritional status, with a reduction in several biochemical nutritional parameters. 

PEW is highly prevalent in CKD patients undergoing dialysis treatment, and its early detection is complex, as biochemical parameters can be affected by uremia restrictions and overhydration. In addition to insufficient nutritional intake, protein loss is associated with dialysis treatments, and inflammation associated with the disease or comorbidities (e.g., cardiovascular disease (CVD), diabetes, or infections) may contribute to malnutrition [9]. According to the guidelines outlined by the National Kidney Foundation’s Kidney Disease Outcomes Quality Initiative, to maintain HD or PD, a minimum of 1.0–1.2 g of protein intake per kg ideal weight per day is required, in conjunction with an adequate energy intake [8].

CKD is a very expensive illness to treat. Early primary and secondary interventions through nutritional advice can prevent and reduce its occurrence and delay the progression of the disease while reducing the risk of developing CVD and other CKD-related complications [16,17]. The lack of adherence to dietary recommendations occurs in 25% to 86% of HD patients and is associated with higher risks of morbidity, mortality, and other symptoms, such as weakness, edema, metabolic complications, nausea, etc. [18,19]. In cases in which dietary advice is not sufficient to meet the nutritional requirements, ONSs are recommended for CKD patients [7,10]. Supplementation with products rich in protein and energy can treat malnutrition in CKD patients, leading to improved nutritional status and decreased morbidity and mortality [7,20]. ONSs can increase daily energy and protein intake up to 10 kcal/kg body weight and 0.3 to 0.4 g/kg body weight, respectively [10]. 

The present study shows the effectiveness of CKD-adapted nutritional supplements since the studied product allowed patients to easily achieve the protein–calorie objectives. The nutritional supplement utilized in the current study is considered a complete and balanced ready-to-drink beverage that is rich in high-quality protein (16 g/bottle) and is highly energetic (2 Kcal/mL). This product is suitable for the dietary management of patients with CKD, and the current results agree with a recent meta-analysis [21], showing that a protein or energy oral supplement can improve the short-term nutritional status of CKD patients receiving dialysis. In the current study, the improvement was associated with an increase in plasma albumin levels and BMI, without influencing blood potassium levels. Other studies have also shown the beneficial effects of nutritional supplements in patients with CKD, suggesting that this treatment helps to achieve nutritional requirements [22]. Indeed, the guidelines of the European Society for Parenteral and Enteral Nutrition (ESPEN) indicate that oral supplementation is the preferred route of refeeding for dialysis patients [7]. BMI is a marker directly associated with nutritional status in patients undergoing HD or PD. Although it has been reported that a high BMI (obesity) in Asian patients receiving PD is associated with a higher risk of mortality (a “V-shaped” relationship) [21], in high BMI patients receiving HD, there is an inverse relationship between these factors [23]. These results, together with the findings of Stenvinkel and Liu et al. [21,23], are in agreement with the current study, which showed an increase of 0.53 kg/m^2^ in the interventional group, compared with the control group, suggesting that oral supplementation improves nutritional status in patients receiving dialysis.

The effectiveness of the dietary supplement was measured according to the updated recommendations for the assessment of nutritional status in CKD patients, including the evaluation of anthropometric parameters (such as triceps fold, circumference, body weight, etc.), biochemical parameters, and dietary assessment [7,24,25]. In line with the nutritional support guidelines for CKD patients, ONS is also enriched with vitamins and minerals, which commonly fall below the desired levels in these patients. Indeed, dialysis patients usually present with significant reductions in a broad range of vitamins, including vitamin D [26]. The current findings showed a significant increase in vitamin D levels in the intervention group. Vitamin C levels were also improved in the intervention group, which may enhance and increase transferrin levels and iron bioavailability to prevent secondary oxalosis [27]. Dialysis patients do not usually show low levels of vitamins B1, B9, B12, or C as these can be replaced orally at the end of the dialysis session. On the other hand, there is increasing evidence of oxidative stress in CKD, especially in patients undergoing dialysis. It is known that selenium plays an important role in the elimination of reactive oxygen species [28]. After 3 months of supplementation with FontActiv^®^ Renal HP, the participants showed significantly increased plasma selenium levels, which could be associated with improvements in other nutritional status measurements. These data are in agreement with the findings reported by Salehi et al. [29], who observed an improvement in malnutrition in HD patients through a significant reduction in oxidative stress and inflammation.

In order to suit the nutritional needs of CKD patients, FontActiv^®^ Renal HP is reduced in sodium, potassium, and phosphorus. Sodium intake should be lower than 3–5 g/day, phosphate intake lower than 1000–1200 mg/day, and potassium intake lower than 2000 mg/day [30]. The content of these elements in FontActiv^®^ Renal HP is very low, with 50.5 mg of sodium and 28 mg of phosphorus and potassium. Some dietary micronutrient deficiencies have been found to contribute to the development of arterial calcifications. Vitamin D plays an important role in maintaining serum calcium within a narrow range and low vitamin D levels have been associated with arterial calcifications [31,32]. In our study, a significant improvement in vitamin D levels was observed only in the intervention arm. Moreover, serum levels of some minerals, such as Mg and Se, have been associated with lower arterial calcification. In the case of Mg, several observational studies suggested that lower serum Mg concentration was significantly associated with arterial calcifications in hemodialyzed patients [33]. In the case of Se, it has been described its potential effect inhibiting arterial calcifications formation and suppressed oxidative [34,35]. In the present study, Se levels significantly increased after the intervention with ONSs, the content of which was 17 µg per bottle in the product.

For nutritional interventions, Hendrick et al. [36] showed that the effectiveness of any intervention depends on the level of compliance. In CKD patients, adherence to a nutritional treatment may be limited by factors such as the length of the dialysis treatment, a lack of social support, etc. Moreover, in nephrology care, noncompliance is characterized by the absence of dialysis sessions, a deliberate shortening of treatment, not following dietary advice (including the amount of fluid recommended), or not following medical prescriptions [37]. Sabatino et al. [10] highlighted the importance of the selection of the ONS to ensure compliance. Acceptability aspects, such as appearance, smell, flavor, taste, preparation, and texture can determine the degree of compliance [10]. In the present study, compliance and acceptance of the flavor (vanilla) and texture were higher than those of other studies. Hernández-Morante et al. [38], using an intervention with Nepro, evaluated the effectiveness of this nutritional supplement in a 4-month prospective study. Of the 60 participants in the nutritional supplement group, 45% abandoned the intervention. Similar rates of compliance were also observed in a European multicenter randomized clinical trial with Renilóm 7.5, in which 47% of participants showed a lack of adherence due to the flavor or excessive satiety [15]. In the current study, the main problem was the tolerance of the supplement. Similar to other studies, supplement intake tolerance was challenging, with a dislike of the supplement flavor, nausea, vomiting, and a feeling of satiety reported after intake [15,39].

The current study has several limitations. First, the sample size was relatively small. In addition, the participants were not adjusted for heterogeneity in terms of ethnicity (most were Caucasian, African, and South American) or other chronic pathologies, which may have affected the results. Additionally, the physical activity and dietary information were self-reported, including the intake of the nutritional supplement. The lack of a biological marker to ensure compliance with the intervention was also a limitation. The major strength of the present study is its RCT design, which allows for an extrapolation of the results. In addition, trained staff collected the data. For the anthropometric measurements, a protocol was established to collect all measurements under the same conditions by the same professional. Moreover, the covariate analysis was detailed, including dietary intake, medication use, sociodemographic characteristics, smoking habit, physical activity levels, and medical history. The multidisciplinary healthcare team was also fully coordinated throughout the intervention.

Due to the necessity to improve adherence, the development of new nutritional supplement formulas adapted to CKD should involve enhancements in a variety of characteristics, including a diversity of flavors, textures, and posology. Moreover, large, randomized clinical trials are necessary to evaluate their efficacies in terms of improvements in quality of life, morbidity, and mortality.

## 5. Conclusions

The current study shows that daily intake of a specific renal nutritional supplement in CKD patients with malnutrition or those at risk of malnutrition may prevent further deterioration in nutritional parameters.

## Figures and Tables

**Figure 1 jcm-11-01647-f001:**
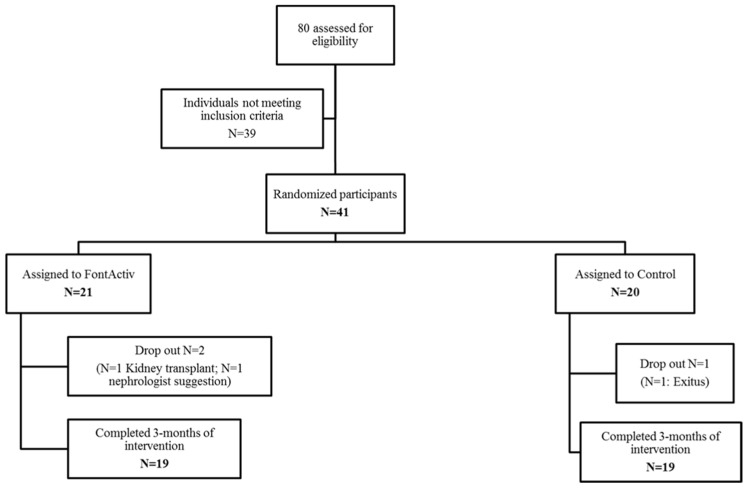
Flowchart of the study.

**Table 1 jcm-11-01647-t001:** Baseline characteristics according to intervention arm.

	Control (n = 19)	FontActiv (n = 19)	*p*-Value
Women, n (%) ^1^	9 (47.4)	11 (57.9)	0.373
Age (years)	76.2 (12.6)	73.6 (17.3)	0.609
Peritoneal dialysis, n (%)	4 (21.0)	3 (15.8)	0.676
Body weight (kg)	65.6 (10.2)	58.3 (10.2)	0.034
BMI (kg/m^2^)	25.3 (3.61)	21.6 (21.6)	0.004
Arm circumference (cm)	28.0 (4.55)	24.1 (2.63)	0.003
Calf circumference (cm)	42.3 (4.51)	38.6 (8.11)	0.106
Triceps fold (mm)	14.0 (5.77)	10.8 (4.31)	0.063
MedDiet P17-score (points)	6.05 (2.25)	5.00 (1.70)	0.112
Prealbumin (g/L)	0.27 (0.08)	0.23 (0.07)	0.095
Albumin (g/L)	37.9 (4.53)	36.5 (4.54)	0.340
Total protein (g/L)	61.9 (6.46)	60.8 (7.52)	0.614
25-hidroxi-vitamin D3 (ng/mL)	15.7 (10.6)	11.6 (6.75)	0.221
HDL-c (mg/dL)	48.1 (11.0)	56.7 (20.0)	0.110
Hemoglobin, g/L	117.9 (11.6)	110.7 (17.9)	0.146
Glycated hemoglobin (%)	6.21 (1.10)	5.35 (0.64)	0.006
Total leukocytes (10^9^/L)	7.14 (2.67)	6.16 (1.50)	0.170
Hematocrit (L/L)	0.37 (0.04)	0.35 (0.05)	0.242
Transferrin saturation	31.5 (15.7)	24.1 (11.4)	0.105
Ferritin (ng/mL)	387.6 (209.4)	506.7 (217.7)	0.094

^1^ Body mass index (BMI), high-density lipoprotein-cholesterol (HDL-c), and 17-point score Mediterranean diet adherence (MedDiet P17-score). Continue variables are expressed as mean (SD). Categorical variables are expressed as number (n) and percentage (%). Comparisons between groups with Pearson’s chi-square test for categorical variables or one-way ANOVA for continuous variables.

**Table 2 jcm-11-01647-t002:** Dietary intake changes (excluding the Oral Supplement) in CKD participants according to the intervention arm after 3 months of intervention.

		Control (n = 19)	FontActiv (n = 19)	
		Mean	Mean	*p*-Value between Groups
Energy (kcal/day) ^1^	Baseline ^1^	2392.1 (404)	2229.2 (454.2)	0.256
	Changes after 3 m.	−164.2 (−335.8, 7.32)	−52.0 (−188.9, 84.8) 3.96)	0.293
Carbohydrates (g/day)	Baseline	233.6 (65.7)	215.1 (47.1)	0.334
	Changes after 3 m.	−21.5 (−47.0, 3.96)	−14.3 (−29.4, 0.76)	0.615
Protein (g/day)	Baseline	96.6 (21.7)	88.5 (17.9)	0.229
	Changes after 3 m.	−6.02 (−13.7, 1.64)	−2.29 (−9.22, 4.63)	0.454
Total fat (g/day)	Baseline	118.6 (20.2)	111.4 (29.9)	0.397
	Changes after 3 m.	−6.16 (−13.1, 0.83)	1.78 (−8.66, 12.2)	0.188
Alcohol (g/day)	Baseline	0.00 [0.00 to 0.09]	0.00 [0.00 to 1.64]	0.916
	Changes after 3 m.	0.00 [0.00 to 0.38]	0.00 [0.00 to 0.00]	0.046
SFAs (g/day)	Baseline	32.2 (8.48)	29.7 (9.95)	0.418
	Changes after 3 m.	−3.29 (−5.82, −0.75) *	0.21 (−2.86, 3.27)	0.071
MUFAs (g/day)	Baseline	60.1 (11.5)	58.7 (15.0)	0.753
	Changes after 3 m.	−1.93 (−4.82, 0.95)	1.84 (−3.89, 7.56)	0.216
PUFAs (g/day)	Baseline	17.4 (4.35)	14.9 (5.18)	0.124
	Changes after 3 m.	−0.38 (−2.05, 1.30)	−0.12 (−1.78, 1.54)	0.823
Cholesterol (mg/day)	Baseline	344.5 (112.3)	304.8 (84.6)	0.234
	Changes after 3 m.	−21.9 (−50.7, 6.86)	1.60 (−18.8, 22.0)	0.174
Sugar (g/day)	Baseline	94.9 (28.3)	86.9 (24.6)	0.369
	Changes after 3 m.	−10.7 (−20.9, −0.57) *	−8.03 (−17.5, 1.43)	0.685
Fiber (g/day)	Baseline	26.7 (8.2)	22.8 (6.1)	0.110
	Changes after 3m.	−1.16 (−2.96, 0.64)	−1.46 (−3.93, 1.00)	0.835
Na (mg/day)	Baseline	3194 (1050)	3064 (1053)	0.710
	Changes after 3 m.	−351.9 (−566.6, −137.3) *	−166.5 (−383.2, 50.2)	0.209
Fe (mg/day)	Baseline	13.9 (3.4)	12.2 (2.7)	0.103
	Changes after 3 m.	−0.58 (−1.62, 0.45)	−0.66 (−1.68, 0.36)	0.918
Zn (mg/day)	Baseline	10.1 (3)	9.4 (2.2)	0.491
	Changes after 3 m.	−0.61 (−1.49, 0.28)	−0.55 (−1.35, 0.26)	0.917
Vitamin A (µg/day)	Baseline	820.2 [630.2 to 1197]	630.1 [545.1 to 826.6]	0.098
	Changes after 3 m.	−7.18 [−42.9 to 15.4]	−13.3 [−61.1 to 16.2]	0.822
Vitamin D (µg/day)	Baseline	4.02 (1.80)	2.93 (1.75)	0.069
	Changes after 3 m.	0.16 (−0.69, 1.00)	0.59 (−0.07, 1.26)	0.403
Vitamin E (mg/day)	Baseline	14.5 (2.74)	12.8 ± 3.92	0.144
	Changes after 3 m.	0.13 (−1.14, 1.39)	0.14 (−1.09, 1.38)	0.988
Vitamin B6 (mg/day)	Baseline	2.25 (0.59)	2.04 (0.67)	0.322
	Changes after 3 m.	−0.13 (−0.28, 0.02)	−0.05 (−0.27, 0.18)	0.514
Vitamin B12 (µg/day)	Baseline	6.66 (2.85)	5.75 (2.98)	0.346
	Changes after 3 m.	0.11 (−0.56, 0.79)	0.15 (−0.51, 0.82)	0.928
Vitamin C (mg/day)	Baseline	189.2 (89.3)	153.6 (73.1)	0.195
	Changes after 3 m.	−33.0 (−63.0, −2.91) *	0.86 (−28.3, 30.0)	0.098
K (mg/day)	Baseline	3564 (1036)	3250 (806.2)	0.312
	Changes after 3 m.	−277.0 (−611.9, 57.9)	−193.4 (−457.0, 70.1)	0.685
Calcium (mg/day)	Baseline	859.9 (254.6)	801.2 (285.2)	0.513
	Changes after 3 m.	−94.7 (−199.9, 10.5)	−28.7 (−112.7, 55.2)	0.313
Folic acid (µg/day)	Baseline	391.3 (142.4)	322.9 (108.2)	0.110
	Changes after 3 m.	−22.0 (−62.7, 18.6)	−19.5 (−51.3, 12.2)	0.920

^1^ Values expressed as mean (SDs), median (IR), and mean differences (95% IC) or median (25th, 75th percentile), as appropriate. Comparisons between groups with one-way ANOVA and comparisons between groups after 3 months of intervention with paired *t*-test. Non-normally distributed variables were compared with Mann–Whitney U test. SFAs, saturated fatty acids; MUFAs, monounsaturated fatty acids; PUFAs, polyunsaturated fatty acids. * Significant differences between baseline and after 3 months (*p* < 0.05).

**Table 3 jcm-11-01647-t003:** Anthropometric changes in CKD participants according to the intervention arm after 3 months of intervention.

		Control (n = 19)	FontActiv (n = 19)	
		Mean	Mean	*p*-Value between Groups
Body weight (kg)	Baseline ^1^	65.6 ± 10.2	58.3 ± 10.2	0.034
	Changes after 3 m.	0.04 (−0.58, 0.66)	1.50 (0.88, 2.12) *	0.002
BMI (kg/m^2^)	Baseline	25.3 ± 3.61	21.6 ± 3.77	0.004
	Changes after 3 m.	0.05 (−0.18, 0.27)	0.54 (0.31, 0.77) *	0.006
Waist circumference (cm)	Baseline	94.5 ± 10.3	82.1 ± 18.3	0.017
	Changes after 3 m.	2.11 (−1.87, 6.08)	1.69 (−2.17, 5.54)	0.883
Hip circumference (cm)	Baseline	99.1 ± 8.20	88.4 ± 17.3	0.026
	Changes after 3 m.	1.96 (−1.01, 4.94)	1.55 (−1.63, 4.73)	0.853
Arm circumference (cm)	Baseline	28.0 ± 4.55	24.1 ± 2.63	0.003
	Changes after 3 m.	−0.48 (−1.35, 0.39)	−0.14 (−0.99, 0.70)	0.595
Calf circumference (cm)	Baseline	42.3 ± 4.51	38.6 ± 8.11	0.106
	Changes after 3 m.	−0.33 (−1.88, 1.22)	1.09 (−0.37, 2.55)	0.190
Triceps fold (mm)	Baseline	14.0 ± 5.77	10.8 ± 4.31	0.063
	Changes after 3 m.	−0.43 (−1.33, 0.47)	0.70 (−0.17, 1.58)	0.079

^1^ Values expressed as mean (SDs). Mean differences (95% IC). Comparisons between groups with one-way ANOVA and comparisons between groups after 3 months of intervention with ANCOVA adjusted for baseline levels of each variable. * Significant differences between baseline and after 3 months (*p* < 0.05).

**Table 4 jcm-11-01647-t004:** Biochemical changes in CKD participants according to the intervention arm after 3 months of intervention.

		Control (n = 19)	FontActiv (n = 19)	
		Mean	Mean	*p*-Value
Prealbumin (g/L) ^1^	Baseline ^1^	0.28 (0.20 to 0.33)	0.22 (0.19 to 0.28)	0.154
	Changes after 3 m.	0.00 (−0.03 to 0.03)	0.02 (−0.03 to 0.05)	0.191
Albumin (g/L)	Baseline	39.0 (36.0 to 41.0)	38.0 (33.0 to 40.0)	0.297
	Changes after 3 m.	0.00 (−2.00 to 1.00)	0.00 (0.00 to 3.00)	0.258
Total protein (g/L)	Baseline	61.9 (6.46)	60.8 (7.52)	0.614
	Changes after 3 m.	0.10 (−2.01 to 2.21)	0.58 (−1.52 to 2.69)	0.743
HDL (mg/dL)	Baseline	47.0 (40.0 to 54.0)	56.0 (40.0 to 66.0)	0.163
	Changes after 3 m.	0.00 (−7.00 to 7.00)	−5.00 (−12.0 to 5.00)	0.271
Hemoglobin (g/L)	Baseline	119.0 (109.0 to 128.0)	112.0 (96.0 to 123.0)	0.172
	Changes after 3 m.	−4.00 (−12.0 to 1.00)	0.00 (−14.0 to 20)	0.246
Glycated hemoglobin (%)	Baseline	5.90 (5.37 to 7.37)	5.30 (4.80 to 5.50)	0.010
	Changes after 3 m.	0.00 (−0.45 to 0.30)	0.00 (−0.52 to 0.30)	0.782
Total leukocytes (10^9^/L)	Baseline	6.44 (5.16 to 7.64)	6.32 (5.00 to 7.24)	0.402
	Changes after 3 m.	0.00 (−1.35 to 0.73)	0.07 (−1.17 to 1.20)	0.795
Hematocrit (L/L)	Baseline	0.37 (0.04)	0.35 (0.05)	0.242
	Changes after 3 m.	−0.01 (−0.03 to 0.00)	0.003 (−0.01 to 0.02)	0.119
Transferrin saturation (%)	Baseline	31.5 (15.7)	24.1 (11.4)	0.105
	Changes after 3 m.	−5.04 (−8.88 to −1.21) *	−3.71 (−7.55 to 0.12)	0.628
Ferritin (ng/mL)	Baseline	393.0 (239.0 to 525.0)	454.0 (376.0 to 564.0)	0.109
	Changes after 3 m.	130.4 (−98.0 to 230.0)	70.0 (−97.0 to 208.0)	0.525
PTH (pg/mL)	Baseline	182.0 (129.0 to 312.0)	212.0 (99.0 to 350.0)	0.832
	Changes after 3 m.	−1.00 (−38.0 to 54.0)	−25.0 (−117.5 to 69.2)	0.711
C-Reactive Protein (mg/dL)	Baseline	0.40 (0.00 to 0.73)	0.58 (0.00 to 1.63)	0.573
	Changes after 3 m.	0.00 (0.00 to 1.16)	0.00 (−1.00 to 0.63)	0.165
Vitamin B1 (ng/mL)	Baseline	66.0 (50.0 to 93.7)	63.5 (54.9 to 120.7)	0.744
	Changes after 3 m.	4.30 (−16.9 to 10.1)	5.40 (−8.20 to 23.7)	0.316
Vitamin B6 (nmol/L)	Baseline	69.1 (21.1 to 101.1)	72.4 (24.1 to 94.3)	0.976
	Changes after 3 m.	−6.80 (−76.3 to 21.6)	9.90 (−18.3 to 59.6)	0.356
Intracellular folic acid (ng/mL)	Baseline	273.5 (74.3)	252.8 (81.0)	0.487
	Changes after 3 m.	66.7 (25.3 to 108.2) *	37.4 (−6.16 to 80.9)	0.326
Serum folic acid (ng/mL)	Baseline	19.6 (6.19 to 24.0)	10.2 (6.26 to 24.0)	0.545
	Changes after 3 m.	−0.11 (−2.00 to 0.06)	0.00 (−1.17 to 1.59)	0.239
Vitamin B12 (pg/mL)	Baseline	380.0 (350.0 to 614.0)	390.0 (313.2 to 517.5)	0.538
	Changes after 3 m.	21.5 (−115.0 to 57.2)	14.5 (−9.00 to 78.7)	0.829
Retinol (µg/dL)	Baseline	72.7 (59.2 to 100.8)	67.5 (55.3 to 86.4)	0.354
	Changes after 3 m.	−7.80 (−23.3 to 10.1)	−2.40 (−10.1 to 8.10)	0.560
25-hydroxyvitamin D(3) (ng/dL)	Baseline	10.5 (8.10 to 22.2)	11.8 (6.40 to 15.6)	0.354
	Changes after 3 m.	3.80 (−5.90 to 4.90)	8.80 (5.80 to 11.4) *	0.016
Alpha-tocoferol (umol/L)	Baseline	32.1 (6.87)	32.6 (7.18)	0.844
	Changes after 3 m.	−3.31 (−6.30 to −0.32) *	−2.33 (−5.10 to 0.44)	0.624
Ca (mg/dL)	Baseline	8.64 (0.78)	8.29 (1.43)	0.360
	Changes after 3 m.	0.17 (−0.18 to 0.52)	0.21 (−0.13 to 0.54)	0.880
P (mg/dL)	Baseline	4.50 (1.75)	4.36 (1.30)	0.771
	Changes after 3 m.	−0.29 (−0.84 to 0.25)	0.49 (−0.04 to 1.02)	0.045
Mg (mg/dL)	Baseline	2.05 (0.34)	2.09 (0.43)	0.779
	Changes after 3 m.	−0.05 (−0.17 to 0.06)	0.02 (−0.11 to 0.16)	0.376
Se (µg/dL)	Baseline	63.5 (16.8)	60.4 (12.3)	0.574
	Changes after 3 m.	4.89 (−3.53 to 13.3)	9.16 (1.37 to 17.0) *	0.450
Zn (µg/dL)	Baseline	77.4 (9.66)	73.0 (17.0)	0.400
	Changes after 3 m.	1.91 (−5.78 to 9.60)	−3.06 (−9.93 to 3.81)	0.333

^1^ Values expressed as mean (SDs), median (IR), and mean differences (95% IC) or median (25th, 75th percentile), as appropriate. Comparisons between groups with one-way ANOVA and comparisons between groups after 3 months of intervention with ANCOVA adjusted for baseline levels of each variable. Non-normally distributed variables were compared with Mann–Whitney U test. PTH, parathormone. * Significant differences between baseline and after 3 months (*p* < 0.05).

## Data Availability

The data presented in this study are available on request from the corresponding author. The data are not publicly available due to ethical concerns.

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
