# Peer review of "A Comparative Study of the Efficacy of an Intervention with a Nutritional Supplement for Patients with Chronic Kidney Disease: A Randomized Trial"

_jcm, 2022, doi:10.3390/jcm11061647_

Round 1
Reviewer 1 Report
The authors state that nutritional supplementation with FontActiv Renal HP improved body weight and BMI of CKD patients. Also vitamin D, selenium and phosphorus status were increased after supplementation with FontActiv Renal HP.
As this is an 'easy to drink' method to improve nutrition status and patients report a good compliance, it would be interesting to see what the effect is of FontActiv Renal HP on a larger cohort of CKD patients. Certainly, as mentioned by the authors, malnutrition contributes to the evolution of CKD progression as well as influences CKD related cardiovascular complications.
However my only concern is that vitamin D and phosphorus were slightly increased. Both are known triggers for the development of arterial calcifications, a major cardiovascular complication in CKD patients. However, on the other hand, Selenium has proven to be protective against the development of arterial calcifications. The authors should include this in the discussion.
Also the authors only mention vitamin D, C and B12. However vitamin K plays an important role in the activation of calcification inhibitors. Moreover CKD patients suffer often from vitamin K deficiency. Did the authors also measure vitamin K levels? I would suggest to include some more information on vitamin K in the manuscript.
Author Response
The authors state that nutritional supplementation with FontActiv Renal HP improved body weight and BMI of CKD patients. Also vitamin D, selenium and phosphorus status were increased after supplementation with FontActiv Renal HP.
As this is an 'easy to drink' method to improve nutrition status and patients report a good compliance, it would be interesting to see what the effect is of FontActiv Renal HP on a larger cohort of CKD patients. Certainly, as mentioned by the authors, malnutrition contributes to the evolution of CKD progression as well as influences CKD related cardiovascular complications.
Answer 1: Thank you for your comments, we absolutely agree with the need to increase the study population in order to confirm the observed results.
However my only concern is that vitamin D and phosphorus were slightly increased. Both are known triggers for the development of arterial calcifications, a major cardiovascular complication in CKD patients. However, on the other hand, Selenium has proven to be protective against the development of arterial calcifications. The authors should include this in the discussion.
Answer 2: Thank you for your comment. While phosphorous levels are significantly increased after the intervention, the levels are normal according to the values established for hemodialyzed patients (Spanish recommendations reference value for P <5 mg/dL) (doi: DOI: 10.3265/Nefrologia.pre2011.Jan.10816). Moreover, PTH and vitamin D are both in normal range according to the reference levels. As we previously mentioned, a large study population is needed in order to confirm the efficacy of this supplement, especially its content in Se and vitamin D.
As you suggested, we included an additional paragraph about arterial calcification (line 390)
“Some dietary micronutrient deficiencies have been found to contribute to the development of arterial calcifications. Vitamin D plays an important role in maintaining serum calcium within a narrow range and low vitamin D levels have been associated with arterial calcifications [31, 32]. In our study, a significant improvement in vitamin D levels was observed only in the intervention arm. Moreover, serum levels of some minerals, such as Mg and Se, have been associated with lower arterial calcification. In the case of Mg, several observational studies suggested that lower serum Mg concentration was significantly associated with arterial calcifications in hemodialyzed patients [33]. In the case of Se, it has been described its potential effect inhibiting arterial calcifications formation and suppressed oxidative [34, 35]. In the present study, Se levels significantly increased after the intervention with the ONS, which content in the product was 17 µg per bottle”
Also the authors only mention vitamin D, C and B12. However vitamin K plays an important role in the activation of calcification inhibitors. Moreover CKD patients suffer often from vitamin K deficiency. Did the authors also measure vitamin K levels? I would suggest to include some more information on vitamin K in the manuscript.
Answer 3: Thank you for your suggestion. Unfortunately, our dietary questionnaire does not include dietary vitamin K intake. The food composition database used for the FFQ does not include this information, being impossible to estimate its intake.
Additionally, no vitamin K plasma levels were available for the present study. Several dietary markers were determined, including, serum and intraerythrocytic folic acid, vitamins A, B1, B12, C, D, and E, zinc, magnesium and selenium. In future studies, we should consider to include Vitamin K parameter as well.

Reviewer 2 Report
The submitted manuscript is very valuable from the point of view of practical implementation of dietary recommendations especially concerning improvement of nutritional status in patients with CKD. As we know malnutrition is a very common problem in this group and its treatment is very challenging due to lack of appetite and dietary restrictions.
The study was well designed. My comments concern :
lack of information about the energy value of the diet and nutrient intake after 3 months of intervention (only baseline values were presented)
a commentary on anthropometric parameters is necessary (in the control group BMI and glycated hemoglobin were statistically significantly higher)
it should be emphasized that it is not known whether the patients in the FontActiv group did not consult a dietician during these 3 months.
Author Response
The submitted manuscript is very valuable from the point of view of practical implementation of dietary recommendations especially concerning improvement of nutritional status in patients with CKD. As we know malnutrition is a very common problem in this group and its treatment is very challenging due to lack of appetite and dietary restrictions.
The study was well designed. My comments concern:
- lack of information about the energy value of the diet and nutrient intake after 3 months of intervention (only baseline values were presented)
Answer 1: Thank you so much for your suggestion. An updated version of the table 2 is available in the new version of the manuscript. In this case, we only represent changes after 3 months of intervention in dietary intake, excluding the use of the oral supplement. No significant differences were observed between groups after the intervention, except for alcohol intake. The results are also included in results section (line 244) and the statistical analysis were described in methods section (line 207).
- a commentary on anthropometric parameters is necessary (in the control group BMI and glycated hemoglobin were statistically significantly higher)
Answer 2: Thank you for your suggestion. We added in the results section the baseline differences observed for both anthropometric and biochemical parameters (line 252 and line 265).
- it should be emphasized that it is not known whether the patients in the FontActiv group did not consult a dietician during these 3 months.
Answer 3: Thank you for your comment. We emphasized this adding “No other dietary intervention were performed in these participants” (line 129)
